# Copper-catalyzed *S*-arylation of Furanose-Fused Oxazolidine-2-thiones

**DOI:** 10.3390/molecules27175597

**Published:** 2022-08-30

**Authors:** Vilija Kederienė, Jolanta Rousseau, Marie Schuler, Algirdas Šačkus, Arnaud Tatibouët

**Affiliations:** 1Department of Organic Chemistry, Kaunas University of Technology, Radvilėnų pl. 19, LT-50254 Kaunas, Lithuania; 2Univ. Artois, CNRS, Centrale Lille, Univ. Lille, UMR 8181–UCCS–Unité de Catalyse et Chimie du Solide, Faculty of Science Jean Perrin, Rue Jean Souvraz SP 18, F-62300 Lens, France; 3Institute de Chimie Organique et Analitique (ICOA), Université d’Orléans, UMR-CNRS 7311, BP 6759, F-45067 Orléans, France

**Keywords:** copper catalysis, carbohydrates, *S*-arylation, C-S bond formation, oxazolidine-2-thione

## Abstract

The 1,3-oxazolidine-2-thiones (OZTs) are important chiral molecules, especially in asymmetric synthesis. These compounds serve as important active units in biologically active compounds. Herein, carbohydrate anchored OZTs were explored to develop a copper-catalyzed C-S bond formation with aryl iodides. Chemoselective *S*-arylation was observed, with copper iodide and dimethylethylenediamine (DMEDA) as the best ligand in dioxane at 60–90 °C. The corresponding chiral oxazolines were obtained in reasonable to good yields under relatively mild reaction conditions. This approach is cheap, as using one of the cheapest transition metals, a simple protocol and various functional group tolerance make it a valuable strategy for getting *S*-substituted furanose-fused OZT. The structures of the novel carbohydrates were confirmed by NMR spectroscopy and an HRMS analysis.

## 1. Introduction

Carbohydrate derivatives are an important class of bioactive compounds with a high density of functional groups [1]. Carbohydrates annulated with heterocycles demonstrate interesting biological properties, including fructose transporter protein GLUT5 inhibition and the selective inhibition of glycoside hydrolase O-GlcNAcase (OGA) or the dual inhibition of OGA and cholinesterases, which are relevant for new therapeutic candidates [2,3,4].

In the past few decades, chemists have paid considerable attention to modifying carbohydrate molecules, which has resulted in the development of innumerable techniques to obtain heteroannulated sugars. Amongst the various reported methods, the most common have used 1,2-annulation strategies such as metal-catalyzed reactions, Michael addition reactions, cycloaddition methods, radical-mediated reactions, or nucleophilic attacks [1]. For example, the formation of 1,2-annulated sugars, having substituted tetrahydropyran and tetrahydrofuran moieties, was investigated via Lewis acid-catalyzed silyl-Prins/alkyne-Prins reactions from appropriately substituted sugar alcohols [5]. Recently, a simple Pd(OAc)_2_-catalyzed strategy was presented using glucose and galactose for the synthesis of sugar-fused indolines via C-H activation/cyclization using the oxalyl group as an auxiliary protecting group [6]. Werz and co-workers explored a convenient approach to obtain carbohydrate-based chromans and isochromans using an intermolecular Pd-catalyzed domino reaction with the catalytic systems Pd(PPh_3_)_4_, [(*t*Bu)_3_PH]BF_4_, and Cs_2_CO_3_ or HN(*i*Pr)_2_, as well as with microwave irradiation [7].

On the other hand, sulfur-containing compounds play an important role in organic synthesis and have been found in various natural products; they also appear in pharmaceuticals, agrochemicals, and functional materials [8,9]. Notably in asymmetric synthesis, a small heterocycle, 1,3-oxazolidine-2-thione (OZT), is a key molecule when chiral. Its parent heterocycle, 1,3-oxazolidine-2-one (OZO), is a chiral template that is well known in the Evans asymmetric methodology and that was specifically modified by Crimmins for the production of acylated OZT [10,11,12,13,14]. This small heterocycle is also well known as a degradation product from natural thioglucosides. The glucosinolates progoitrin, epiprogoitrin, and glucobarbarin are chiral OZTs resulting from the glucohydrolase action of a specific enzyme, myrosinase [15,16,17,18,19,20,21]. These molecules, as part of the defense mechanism of plants, have been shown to have several biological impacts—most notably on feedstock, but also on human health as inhibitors of tyrosine kinase [22,23]. They are also known for various biological activities such as anticancer, antibacterial, antifertility, and insecticidal activities [24]. A few representative biologically relevant OZT-based molecules are shown in Figure 1.

Chemists have developed different methods of synthesis for OZT or have analyzed its potential as an asymmetric inducer [24,28,29,30,31,32,33]. OZT connected to a carbohydrate backbone has been the subject of various studies to control the geometry of the carbohydrate structure using the specific chemistry of such heterocycles as well as to access new biologically active compounds [34]. The reactivity of OZT anchored on carbohydrates has been studied through classical reactions such as acylation, sulfonylation, and alkylation, as well as oxidation. OZT connected to carbohydrate templates were studied as precursors of 2-alkyl or 2-aryl 1,3-oxazoline following a copper-promoted palladium-catalyzed cross-coupling reaction called the Liebeskind–Srogl reaction, which uses a range of organoboronyl and organostannyl reagents [35,36,37]. Recently, carbohydrate-anchored OZTs were explored for the formation of iminosugars as inhibitors of glycosidases [38].

Herein, we present our preliminary results on the exploration of a chemoselective copper-catalyzed *S*-arylation reaction of aryl iodides with these unusual chiral oxazolidine-2-thiones anchored onto carbohydrate backbones. Copper-catalyzed *S*-arylation is a well-known process, but to our knowledge, it has not been applied to chiral OZT, or connected to carbohydrate backbones, in which the metal-catalyzed C-S bond formation has been scarcely explored [39,40,41]. Only a few examples have described the synthesis of *S*-arylated oxazolidinethiones, either through a one-pot procedure using isocyanide dichloride in a two-step approach [42] or using the reactivity of transient arynes with 2-oxazolidinethiones to obtain selective *S*-arylation [43].

## 2. Results and Discussion

The oxazolidine-2-thione moiety has two potential nucleophilic sites: sulfur and nitrogen atoms. Accordingly, both *S*-arylation and *N*-arylation may take place due to the existence of two tautomeric thione–thiol forms (Figure 1). Sulfur, as a soft nucleophile, can prevail in the formation of *S*-arylated products in a cross-coupling reaction [44,45,46].

D-arabinofuranose-, D-xylofuranose-, and D-ribofuranose-derived oxazolidine-2-thiones **1**–**3** can easily be obtained from the corresponding carbohydrates using potassium thiocyanate under acidic conditions [2,36,47]. D-arabinose was chosen as the starting precursor in our study [48]. The protection of the hydroxyl groups was achieved with acetyl (Ac) or *tert*-butyldimethylsilyl (TBDMS) groups, allowing the selective protection of the alcohols obtained without modifying the oxazolidinethione group [47,49,50] (Figure 2). The reaction of D-arabinose OZT **1** with acetic anhydride in pyridine resulted in the pre-acetylated product-derivative **4** in a quantitative yield without any requirement for purification. The crude product was selectively deacetylated in a mixture of methanol and pyridine. These conditions resulted in an efficient selective deacetylation of the acyl-protected 1,3-oxazolidine-2-thione vs. the ester group. The silylation process directly produced a selective protection of the hydroxyl groups of the three pentose OZTs **1**, **2**, and **3.** The corresponding silyl- protected OZTs (**6**–**8**) were obtained in very good yields of 96%, 98%, and 95%, respectively.

We decided to convert OZT **5** into the corresponding oxazolidin-2-one (OZO) derivative **10** (Figure 3), which could then be tested with a Cu-catalyzed cross-coupling reaction. We have previously reported that the direct conversion of OZT derivatives to OZO derivatives by oxidation is not efficient [2]. Prior to oxidation, OZT template **5** required *S*-alkylation and was converted into the corresponding 2-benzylsulfanyloxazoline **9** at a 92% yield. After oxidation, OZO **10** was obtained at a 57% yield.

Copper-catalyzed coupling reactions was performed by applying standard conditions with various copper ligands and copper iodide, which was chosen for its air stability [51,52,53]. We first chose the D-arabino derivative **5** with 4-iodoanisole as a model substrate to optimize the catalytic *S*-arylation (Figure 4, Table 1). As a control experiment (Table 1, entry 1), no product was observed if the reaction conditions were carried out in the absence of the copper ligand. When the reaction was performed with 20 mol% of CuI and 40 mol% of dimethylethylenediamine (DMEDA) under basic conditions (Cs_2_CO_3_), C-S coupling was observed with a better, but lower, yield of 20% (Table 1, entry 3). A reduced or increased ratio of CuI and DMEDA did not improve the yields. A reduction in the temperature to 60 °C maintained the results at 35% (Table 1, entry 5). Thus, the 1:2 ratio was maintained and proved to be similar to other copper-catalyzed cross-coupling reactions [54].

Applying these conditions (Table 1, entry 6) to the silyl-protected OZT **6** resulted in a low yield of 22%. Pushing the conditions with a higher temperature (Table 1, entry 7) gave a much better yield of 69%. The ligands, including *L*-proline, DABCO, 2-aminopyridine, ethyl 2-oxocyclohexane-1-carboxylate, and 9,10-phenanthroline, were tested to enhance the C-S coupling reaction between OZT **6** and 4-iodoanisole using the conditions of entry 7 (Table 1, entries 8–13). None of them were effective for this coupling reaction. The coupling reaction with ethyl 2-oxocyclohexane-1-carboxylate **L5** and DABCO **L7** did not perform at all (Table 1, entries 11 and 13); in other cases, a degradation of the starting compound was observed (Table 1, entries 1, 8, 9 and 12). Only the 9,10-phenanthroline **L4** showed reactivity, but its efficiency was worse (Table 1, entry 10). Of all these ligands, only DMEDA **L1** was efficient in the C-S coupling reaction. As expected with benzylsulfanyloxazoline 9 and oxazolidinone 10, no coupling reaction was observed (Table 1, entries 14–15).

Notably, only the *S*-arylation was observed; no *N*-arylation was detected, although the *N*-arylation of aryl halides has been shown to proceed under similar conditions [55,56,57]. In this case, *S*-arylation was preferred compared with *N*-arylation. Sulfur, as a soft nucleophile, favors an interaction with copper and occurs in the formation of S-arylated products in a cross-coupling reaction [44]. Furthermore, this can be explained by the mechanism of the copper-assisted nucleophilic substitution reaction [58]. In the intermediate catalysis step, the mercapto group mainly attacks the copper complex, which is a soft metal atom. Sulfur is a stronger nucleophile than nitrogen due to its nature of high polarizability, a large size, and more electron lone pairs [58].

Although the exact mechanism remains unknown for this Cu-catalyzed *S*-arylation process, there are valuable experimental and theoretical mechanistic studies of Cu-catalysed *N*-arylation [59,60,61]. Moreover, there are investigations that the formation of the Cu(I)-complex nucleophile and the coupling product depends on the concentration of the ligand [62,63]. Applying optimized reaction conditions (Table 1, entry 7), the *S*-arylation reaction of **6** was explored with various functionalized aryl iodides (Figure 5). The results are summarized in Table 2. Aryl iodides were chosen for their enhanced reactivity compared with other aryl halides. As shown in Table 2, D-arabinose derivative **6** reacted with various substituted aryl iodides, leading to the corresponding products **12**–**19** with low to good yields (Figure 2).

The aryl iodides with electron-donating groups (such as the methoxy group) and electron-withdrawing groups (such as the nitro group) obtained the desired *S*-aryl oxazolines in moderate to good yields (Table 2, entries 1–6). No clear evidence of the electronic effect could be extracted from these results. Whether the substitution of iodobenzene was at the para-, meta-, or ortho-position, the C–S bond formation resulted in moderate yields and no effect of steric hindrance was clearly visible. On the contrary, the comparison of the ortho-substitution with ortho-fluorobenzene and ortho-trifluoromethyl benzene showed a significant drop in yields. The relatively poor yields obtained by performing the coupling reaction with ortho-substituted iodobenzene containing CF_3_ and F groups could be attributed to the instability of the compounds (Table 2, entries 7–8). Degradation could have occurred through the hydrolysis of phenylsulfanyloxazoline. When left under basic conditions, the formation of the corresponding oxazolidinone was detected.

The application of this cross-coupling reaction on D-xylose oxazolidine-2-thione **7** and D-ribose oxazolidine-2-thione **8** was explored using ortho-substituted iodobenzene derivatives (Figure 6).

Both compounds proved that the method was effective on other carbohydrate anchored 1,3-oxazolidine-2-thiones but with reduced yields. The derivatives D-xylo **20**–**21** and D-ribo **22** were obtained with 58%, 45%, and 46% yields, respectively.

## 3. Materials and Methods

### 3.1. General Information

Reactions in anhydrous conditions were performed under an argon atmosphere in pre-dried flasks, using anhydrous solvents (distilled, when necessary, according to D. D. Perrin, W. L. F. Armarego and D. R. Perrin in *Purification of Laboratory Chemicals*, Pergamon, Oxford, 1986). Molecular sieves were activated prior to use by heating for 4 h at 500 °C. All reagents were obtained from commercial chemical suppliers and used without further purification. The course of the reactions was monitored by an initial TLC analysis on precoated aluminum foil-backed plates (Merck Kieselgel 60 F_254_, Darmstadt, Germany). TLC results were visualized using standard visualization techniques or agents: UV fluorescence (254 nm) and staining with a 1% aq potassium permanganate solution or a heat treatment with a 10/85/5 mixture of sulfuric acid/ethanol/water. Flash column chromatography was performed with Silica Gel 60 Å (230–400 µm, Merck KGaA, Darmstadt, Germany). The melting points (°C) were measured with a Thermo Scientific 9200 capillary apparatus and are uncorrected. The NMR spectra were recorded on a 400 MHz Bruker Avance 2 spectrometer (Bruker BioSpin AG, Fallanden, Switzerland) (400 MHz (^1^H), 100 MHz (^13^C), and 376 MHz (^19^F)). The chemical shifts are expressed in parts per million (ppm) downfield from the TMS internal standard. The coupling patterns for ^1^H NMR are designated as s = singlet, d = doublet, t = triplet, and m = multiplet, (a few ^13^C NMR are designated as d = doublet, and q = quartet); the coupling constants are given in Hz. The NMR peak assignments were elucidated via DEPT, COSY, and HSQC techniques for all reported compounds. The IR spectra from the samples in a neat form were measured with a Thermo Scientific Nicolet iS10 FT-IR spectrophotometer. IR absorption frequencies are given in cm^–1^. Optical rotations were measured at 20 °C with a Perkin Elmer 341 polarimeter and were given in g^–1^⋅cm^3^⋅dm^–1^. Low-resolution mass spectra (MS) were recorded with a Perkin–Elmer Sciex API 300 spectrometer [ionspray (IS) mode] (PerkinElmer Inc., Waltham, MA, USA). High-resolution mass spectra (HRMS) were recorded with a Bruker MaXis spectrometer [electrospray ionization (ESI) mode] (Bruker DaltonikGmbH, Bremen, Germany). ^1^H, ^13^C NMR spectra and HRMS data of all new compounds are provided in Appendix A.

### 3.2. Synthesis of N-acetyl-4,5-dihydro(3′,5′-di-O-acetyl-1′,2′-dideoxy-β-D-arabinofuranoso)-[1,2-d]-oxazolidine-2-thione (**4**)

To a solution of compound **1** (500 mg, 2.61 mmol) in dry pyridine (8 mL), acetic anhydride (2 mL) was added dropwise to a well-stirred solution at 0 °C under argon for 10 min. Then the resulting reaction mixture was stirred at room temperature for 1 h. The reaction mixture was diluted with ethyl acetate and washed with 10% acetic acid, then water and sat. aq. NaHCO_3_ solution, and finally dried over MgSO_4_. After filtration, the solvent was removed under vacuum. The obtained compound **4** was a colorless oil of very good purity (830 mg, quantitative yield); ${\left[\alpha \right]}_{D}^{20}$ = −69 (*c* = 0.5, CHCl_3_). ^1^H NMR (400 MHz, CDCl_3_): *δ* = 2.08 (s, 3H, CH_3_), 2.12 (s, 3H, CH_3_), 2.79 (s, 3H, *C*H_3_(C=O)N), 4.03 (dd, 1H, *J*_5′b,4′_ = 4.6 Hz, *J*_5′b,5′a_ = 11.9 Hz, 5′b-H), 4.30 (dd, 1H, *J*_5′a,4′_ = 5.8 Hz, *J*_5′a,5′b_ = 11.9 Hz, 5′a-H), 4.35−4.38 (m, 1H, 4′-H), 5.06 (d, 1H, *J*_2′,1′_ = 5.6 Hz, 2′-H), 5.32 (d, 1H, *J*_3′,4′_ = 1.0 Hz, 3′-H), 6.50 (d, 1H, *J*_1′,2′_ = 5.6 Hz, 1′-H) ppm. ^13^C NMR (100 MHz, CDCl_3_): *δ* = 20.8 (CH_3_), 20.9 (CH_3_), 26.2 (*C*H_3_(C=O)N), 63.1 (C-5′), 77.0 (C-3′), 84.0 (C-4′), 85.5 (C-1′), 90.9 (C-2′), 169.7 (C=O), 170.7 (C=O), 170.8 (C=O), 183.7 (C=S) ppm. IR (NEAT): *ν* = 1740, 1716 (C=O), 1366, 1324, 1214, 1162 (C-O, C-N) cm^−1^. MS (IS): *m/z* = 318 [M + H]^+^, 340 [M + Na]^+^. HRMS (ESI): *m*/*z* [M + H]^+^ calcd. for C_12_H_16_NO_7_S: 318.06420; found: 318.06432. HRMS (ESI): *m*/*z* [M + Na]^+^ calcd. for C_12_H_15_NNaO_7_S: 340.04614; found: 340.04654.

### 3.3. Synthesis of 4,5-dihydro(3′,5′-di-O-acetyl-1′,2′-dideoxy-β-D-arabinofuranoso)-[1,2-d]-oxazolidine-2-thione (**5**)

Under argon, to a cooled solution of compound **4** (830 mg, 2.61 mmol) in dry pyridine (4 mL) at 0 °C, methanol (1 mL) was added dropwise. The reaction mixture was stirred at 0 °C for 15 min, then was allowed to reach room temperature and stirred for 48 h. The reaction mixture was diluted with ethyl acetate (60 mL) and washed with 10% cold acetic acid solution (20 mL), cold water (20 mL), and then sat. aq. NaCl solution (20 mL), and finally dried over MgSO_4_. After filtration, the solvent was removed under reduced pressure. The residue was purified by flash chromatography (eluent: PE/EtOAc, 1:1, *R_f_* = 0.27) to give **4** (689 mg, 96%) as a white solid; mp 157−159 °C, ${\left[\alpha \right]}_{D}^{20}$ = −34 (*c* = 0.49, CHCl_3_). ^1^H NMR (400 MHz, DMSO-*d_6_*): *δ* = 2.06 (s, 3H, CH_3_), 2.08 (s, 3H, CH_3_), 3.96 (dd, 1H, *J*_5__′b,4__′_ = 6.5 Hz, *J*_5__′b,5__′a_ = 11.9 Hz, 5′b-H), 4.06 (dd, 1H, *J*_5__′a,4__′_ = 5.1 Hz, *J*_5__′a,5__′b_ = 11.9 Hz, 5′a-H), 4.31−4.34 (m, 1H, H-4′), 5.20 (s, 1H, H-3′), 5.37 (d, 1H, *J*_2__′,1__′_ = 5.7 Hz, H-2′), 5.91 (d, 1H, *J*_1__′,2__′_ = 5.7 Hz, H-1′), 11.06 (s, 1H, NH). ^13^C NMR (100 MHz, DMSO-*d_6_*): *δ* = 20.5 (CH_3_), 20.6 (CH_3_), 62.9 (C-5′), 76.7 (C-3′), 81.3 (C-4′), 88.7 (C-1′), 89.9 (C-2′), 169.5 (C=O), 170.1 (C=O), 188.0 (C=S). IR (NEAT): *ν* = 3263 (N-H), 1746, 1704 (C=O), 1503, 1313, 1212, 1165, 1048 (C-O, C-N) cm^−1^. MS (IS): *m/z* = 276 [M + H]^+^, 298 [M + Na]^+^. HRMS (ESI): *m*/*z* [M + H]^+^ calcd. for C_10_H_14_NO_6_S: 276.05363; found: 276.05369. HRMS (ESI): *m*/*z* [M + Na]^+^ calcd. for C_10_H_13_NNaO_6_S: 298.03558; found: 298.03584.

### 3.4. Synthesis of 2-benzylsulfanyl-4,5-dihydro(3′,5′-di-O-acetyl-1′,2′-dideoxy-β-D-arabinofuranoso)-[1,2-d]-oxazole (**9**)

To a solution of compound **5** (0.3 g, 1.09 mmol) in dry dichloromethane (5 mL), triethylamine (0.46 mL, 0.73 mmol) and benzyl bromide (0.26 mL, 2.18 mmol) were added dropwise. The reaction mixture was stirred at room temperature under argon for 24 h. It was then diluted with dichloromethane (60 mL) and washed with 1M HCl (20 mL), NaHCO_3_ sat. (20 mL), and NaCl sat. (20 mL) solutions, and finally dried over MgSO_4_. After filtration, the solvent was evaporated in vacuo. The obtained residue was purified by flash chromatography (eluent: PE/EtOAc, 7:1, *R_f_* = 0.26) to give **5** (368 mg, 92%) as colorless oil; ${\left[\alpha \right]}_{D}^{20}$ = −69 (*c* = 1, CHCl_3_). ^1^H NMR (400 MHz, CDCl_3_): *δ* = 2.07 (s, 3H, CH_3_), 2.10 (s, 3H, CH_3_), 3.94 (dd, 1H, *J*_5__′b,4__′_ = 6.3 Hz, *J*_5__′b,5__′a_ = 11.7 Hz, 5′b-H), 4.05 (dd, 1H, *J*_5__′a,4__′_ = 6.5 Hz, *J*_5__′a,5__′b_ = 11.7 Hz, 5′a-H), 4.26 (dd, 1H, *J*_4__′,3__′_ = 2.0 Hz, *J*_4__′,5__′b_ = 6.3 Hz, H-4′), 4.30 (s, 2H, SCH_2_), 4.94 (d, 1H, *J*_2__′,1__′_ = 5.9 Hz, H-2′), 5.18 (d, 1H, *J* = 2,0 Hz, H-3′). 6.12 (d, 1H, *J*_1__′,2__′_ = 5.9 Hz, H-1′), 7,26−7,33 (m, 3H, H_Ar_); 7,37−7,39 (m, 2H, H_Ar_) ppm. ^13^C NMR (100 MHz, CDCl_3_): *δ* = 20.8 (2×CH_3_), 36.5 (SCH_2_), 63.2 (C-5′), 78.2 (C-3′), 81.7 (C-4′), 87.7 (C-2′), 101.4 (C-1′), 127.8 (CH), 128.7 (2×CH), 129.1 (2×CH), 136.1 (C_q_), 169.8 (C_q_), 170.2 (C=O), 170.5 (C=O) ppm. IR (NEAT): *ν* = 2959, 1740 (C=O), 1596, 1366, 1212, 1134, 1029 (C-O, C-N) cm^−1^. MS (IS): *m/z* = 366 [M + H]^+^, 388 [M + Na]^+^. HRMS (ESI): *m*/*z* [M + H]^+^ calcd. for C_17_H_20_NO_6_S: 366,10058; found: 366,10104. HRMS (ESI): *m*/*z* [M + Na]^+^ calcd. for C_17_H_19_NNaO_6_S: 388,08253; found: 388,08296.

### 3.5. Synthesis of 4,5-dihydro(3′,5′-di-O-acetyl-1′,2′-dideoxy-β-D-arabinofuranoso)-[1,2-d]-oxazolidine-2-one (**10**)

To a solution of compound **9** (270 mg, 0.75 mmol) in dry dichloromethane (5 mL), dry NaHCO_3_ (188 mg, 2.24 mmol) was added. Then, the reaction mixture was cooled to 0 °C temperature under argon for 10 min, then *m*-CPBA (75%, 490 mg, 2.84 mmol) was added slowly. The resulting reaction mixture was stirred at 0 °C for 2 h. Then the reaction mixture was diluted with dichloromethane and washed with sat. aq. Na_2_S_2_O_5_ solution, then sat. aq. NaHCO_3_ solution, and finally dried over MgSO_4_. After filtration, the solvent was evaporated in vacuo. The obtained residue was purified by flash chromatography (eluent: petroleum ether/EtOAc, 4:6, *R_f_* = 0.22) to give **10** (110 mg, 57%) as white solid; mp 115–116 °C; ${\left[\alpha \right]}_{D}^{20}$ = −63 (*c* = 1.0, CHCl_3_). ^1^H NMR (400 MHz, CDCl_3_): *δ* = 2.09 (s, 3H, CH_3_), 2.10 (s, 3H, CH_3_), 3.96 (dd, 1H, *J*_5__′b,4__′_ = 2.9 Hz, *J*_5__′b,5__′a_ = 10.4 Hz, 5′b-H), 4.25−4.31 (m, 2H, 5′a-H, 4′-H), 4.98 (d, 1H, *J*_2__′,1__′_ = 5.7 Hz, 2′-H), 5.25 (s, 1H, 3′-H), 5.77 (d, 1H, *J*_1__′,2__′_ = 5.7 Hz, 1′-H), 6.45 (br s, 1H, NH) ppm. ^13^C NMR (100 MHz, CDCl_3_): *δ* = 20.7 (CH_3_), 20.9 (CH_3_), 63.8 (C-5′), 78.0 (C-3′), 83.3 (C-4′), 84.4 (C-2′), 87.3 (C-1′), 156.8 (C=O), 169.9 (C=O), 171.0 (C=O) ppm. IR (NEAT): *ν* = 3316 (N-H), 1788, 1749, 1711 (C=O), 1377, 1267, 1223, 1212, 1092, 1051 (C-O, C-N) cm^−1^. MS (IS): *m/z* = 260 [M + H]^+^; 282 [M + Na]^+^. HRMS (ESI): *m*/*z* [M + H]^+^ calcd. for C_10_H_14_NO_7_: 260.07648; found: 260.07669. HRMS (ESI): *m*/*z* [M + Na]^+^ calcd. for C_10_H_13_NNaO_7_: 282.05842; found: 282.05856.

### 3.6. Synthesis of 4,5-dihydro(3′,5′-di-O-tert-butyldimethylsilyl-1′,2′-dideoxy-β-D-arabinofuranoso)-[1,2-d]-oxazolidine-2-thione (**6**)

Compound **1** (400 mg, 2.08 mmol) was dissolved in dry DMF (5 mL). Imidazole (712 mg, 10.47 mmol) and *tert*-butyldimethylsilyl chloride (792 mg, 2.52 mmol) were then added at 0 °C temperature. Then, the reaction mixture was stirred at room temperature for 15 h. The reaction mixture was diluted with dichloromethane (100 mL) and washed three times with water (3 × 30 mL), then brine (30 mL), and finally dried over MgSO_4_. After filtration, the solvent was removed by evaporation in vacuo. The obtained residue was purified by flash chromatography (eluent: petroleum ether/EtOAc, 95:5 after 9:1, *R_f_* = 0.2) to give **6** (841 mg, 96%) as white solid; mp 64–65 °C; ${\left[\alpha \right]}_{D}^{20}$ = −52 (*c* = 1.1, CHCl_3_). ^1^H NMR (400 MHz, CDCl_3_): *δ* = 0.05 (s, 3H, CH_3_Si), 0.06 (s, 3H, CH_3_Si), 0.11 (s, 3H, CH_3_Si), 0.12 (s, 3H, CH_3_Si), 0.87 (s, 9H, 3 × CH_3_), 0.88 (s, 9H, 3×CH_3_), 3.40−3.45 (m, 1H, 5′b-H), 3.63 (dd, 1H, *J*_5__′a,4__′_ = 5.5 Hz, *J*_5__′a,5__′b_ = 10.5 Hz, 5′a-H), 4.04−4.07 (m, 1H, 4′-H), 4.53 (s, 1H, 3′-H), 5.01 (d, 1H, *J*_2__′,1__′_ = 5.6 Hz, 2′-H), 5.82 (d, 1H, *J*_1__′,2__′_ = 5.6 Hz, 1′-H), 7.40 (br s, 1H, NH) ppm. ^13^C NMR (100 MHz, CDCl_3_): *δ* = -5.2 (CH_3_Si), -5.1 (CH_3_Si), -4.7 (2×CH_3_Si), 18.2 (C_q_), 18.5 (C_q_), 25.9 (3 × CH_3_), 26.1 (3 × CH_3_), 62.5 (C-5′), 76.2 (C-3′), 88.2 (C-4′), 89.5 (C-1′), 93.2 (C-2′), 189.0 (C=S) ppm. MS (IS): *m/z* = 420 [M + H]^+^. HRMS (ESI): *m*/*z* [M + Na]^+^ calcd. for C_18_H_37_ NNaO_4_SSi_2_: 442.18740; found: 442.18742. 

### 3.7. Synthesis of 4,5-dihydro(3′,5′-di-O-tert-butyldimethylsilyl-1′,2′-dideoxy-α-D-xylofuranoso)-[1,2-d]-oxazolidine-2-thione (**7**)

Compound **2** (600 mg, 3.14 mmol) was dissolved in dry DMF (6 mL). Imidazole (1079 mg, 15.85 mmol) and *tret*-butyldimethylsilyl chloride (1192 mg, 7.90 mmol) were then added at 0 °C temperature. Then, the reaction mixture was stirred at room temperature for 24 h. The reaction mixture was diluted with dichloromethane and washed with water, then brine, and finally dried over MgSO_4_. After filtration, the solvent was removed by evaporation in vacuo. The obtained residue was purified by flash chromatography (eluent: petroleum ether/EtOAc, 95:5 after 9:1, *R_f_* = 0.35) to afford **7** (1.3 g, 98%) as white solid; mp 106−107 °C; ${\left[\alpha \right]}_{D}^{20}$ = −22 (*c* = 1.0, CHCl_3_). ^1^H NMR (400 MHz, CDCl_3_): *δ* = 0.03 (s, 3H, CH_3_Si), 0.04 (s, 3H, CH_3_Si), 0.09 (s, 3H, CH_3_Si), 0.11 (s, 3H, CH_3_Si), 0.86(s, 9H, 3 × CH_3_), 0.86 (s, 9H, 3 × CH_3_), 3.74−3.81 (m, 2H, 5′a,b-H), 3.94 (dt, 1H, *J*_4__′,5__′a_ = 2.9 Hz, *J*_4__′,5__′b_ = 8.7 Hz, 4′-H), 4.40 (d, 1H, *J*_3__′,4__′_ = 2.6 Hz, 3′-H), 4.95 (d, 1H, *J*_2__′,1__′_ = 5.4 Hz, 2′-H), 5.83 (d, 1H, *J*_1__′,2__′_ = 5.4 Hz, 1′-H), 7.86 (s, 1H, NH) ppm. ^13^C NMR (100 MHz, CDCl_3_): *δ* = -5.2 (CH_3_Si), -5.2 (CH_3_Si), -5.0 (CH_3_Si), -4.7 (CH_3_Si), 18.2 (C_q_), 18.5 (C_q_), 25.8 (3 × CH_3_), 26.1 (3 × CH_3_), 60.3 (C-5′), 74.4 (C-3′), 81.3 (C-4′), 88.6 (C-1′), 91.2 (C-2′), 189.4 (C=S) ppm. IR (NEAT): *ν* = 3286 (N-H), 2954, 2929 (C-H_Al_), 1502, 1254, 1153, 1003, 833 (C-O, C-N) cm^−1^. MS (IS): *m/z* = 420 [M + H]^+^. HRMS (ESI): *m*/*z* [M + H]^+^ calcd. for C_18_H_38_NO_4_SSi_2_: 420.20546; found: 420.0565. 

### 3.8. General Procedure: Copper-catalyzed S-arylation of Compounds **11**–**21**

One of the appropriate starting oxazolidine-2-thiones **5**, **6**, **7** or **8** (1 equiv), Cs_2_CO_3_ (2 equiv), the copper iodide (0.2 equiv), and the iodide derivative (1.5 equiv) were dissolved in anhydrous dioxane (2 mL) in a round-bottom flask under argon. After 10 min, DMEDA (0.4 equiv) was added dropwise and the reaction was stirred at 60 °C or 90 °C for 24 h. The reaction was then allowed to cool to room temperature, and poured into a sat. aq NaCl (40 mL) solution and extracted with EtOAc (2 × 25 mL), then with H_2_O (1 × 10 mL), and dried over MgSO_4_. After the evaporation of the solvent, the residue was purified by flash chromatography using PE/EtOAc as an eluent to produce the desired products. The data for the selected compounds are described below.

#### 3.8.1. 2-[(4-Methoxyphenyl)sulfanyl]-4,5-dihydro(3′,5′-di-O-acetyl-1′,2′-dideoxy-β-D-arabinofuranoso)-[1,2-d]-oxazole (**11**)

Prepared from **5** (100 mg, 0.36 mmol) and 4-iodoanisole (127 mg, 0.54 mmol), the obtained residue was purified by flash chromatography (eluent: petroleum ether/EtOAc, 1:1, *R_f_* = 0.27) to give **11** (48 mg, 35%) as a colorless oil; ${\left[\alpha \right]}_{D}^{20}$ = −64 (*c* = 1.2, CHCl_3_). ^1^H NMR (400 MHz, CDCl_3_): *δ* = 2.07 (s, 3H, CH_3_), 2.10 (s, 3H, CH_3_), 3.80 (s, 3H, OCH_3_), 4.03 (dd, 1H, *J*_5__′b,4__′_ = 7.0 Hz, *J*_5__′b,5__′a_ = 11.6 Hz, 5′b-H), 4.10 (dd, 1H, *J*_5__′a,4__′_ = 6.4 Hz, *J*_5__′a,5__′b_ = 11.6 Hz, 5′a-H), 4.23–4.26 (m, 1H, 4′-H), 4.90 (d, 1H, *J*_2__′,1__′_ = 5.8 Hz, 2′-H), 5.19 (s, 1H, 3′-H), 6.05 (d, 1H, *J*_1__′,2__′_ = 5.8 Hz, 1′-H), 6.90 (d, 2H, *J*_3,2_ = *J*_5,6_ = 8.8 Hz, 3-H_Ph_, 5-H_Ph_), 7.49 (d, 2H, *J*_2,3_ = *J*_6,5_ = 8.8 Hz, 2-H_Ph_, 6-H_Ph_) ppm. ^13^C NMR (100 MHz, CDCl_3_): *δ* = 20.9 (CH_3_), 21.0 (CH_3_), 55.6 (OCH_3_), 63.4 (C-5′), 78.3 (C-3′), 81.8 (C-4′), 87.7 (C-2′), 101.8 (C-1′), 115.3 (2 × C_Ph_), 116.9 (C_q_), 127.1 (C_q_), 137.0 (2 × C_Ph_), 161.4 (C-S), 169.9 (C=O), 170.8 (C=O) ppm. IR (NEAT): *ν* = 1740 (C=O), 1592, 1494, 1215, 1129, 1025 (C-O, C-N, C=C, C=N) cm^−1^. MS (IS): *m/z* = 382 [M + H]^+^. HRMS (ESI): *m*/*z* [M + H]^+^ calcd. for C_17_H_20_NO_7_S: 382.09550; found: 382.09618. HRMS (ESI): *m*/*z* [M + Na]^+^ calcd. for C_17_H_19_NNaO_7_S: 404.07744; found: 404.07785.

#### 3.8.2. 2-[(4-Methoxyphenyl)sulfanyl]-4,5-dihydro(3′,5′-di-O-tert-butyldimethylsilyl-1′,2′-dideoxy-β-D-arabinofuranoso)-[1,2-d]-oxazole (**12**)

Prepared from **6** (100 mg, 0.24 mmol) and 4-iodoanisole (84 mg, 0.36 mmol), the obtained residue was purified by flash chromatography (eluent: petroleum ether/EtOAc, 9:1, *R_f_* = 0.18) to give **12** (86 mg, 69%) as a colorless oil; ${\left[\alpha \right]}_{D}^{20}$ = −67 (*c* = 0.5, CHCl_3_). ^1^H NMR (400 MHz, CDCl_3_): *δ* = 0.06 (s, 3H, CH_3_Si), 0.07 (s, 6H, 2 × CH_3_Si), 0.08 (s, 3H, CH_3_Si), 0.86 (s, 9H, 3 × CH_3_), 0.90 (s, 9H, 3 × CH_3_), 3.42 (dd, 1H, *J*_5__′b,4__′_ = 8.5 Hz, *J*_5__′b,5__′a_ = 10.5 Hz, 5′b-H), 3.63 (dd, 1H, *J*_5__′a,4__′_ = 4.6 Hz, *J*_5__′a,5__′b_ = 10.6 Hz, 5′a-H), 3.79 (s, 3H, OCH_3_), 3.87−3.91 (m, 1H, 4′-H), 4.37 (br s, 1H, 3′-H), 4.71 (d, 1H, *J*_2__′,1__′_ = 5.9 Hz, 2′-H), 5.96 (d, 1H, *J*_1__′,2__′_ = 5.9 Hz, 1′-H), 6.89 (d, 2H, *J*_3,2_ = *J*_5,6_ = 8.6 Hz, 3-H_Ph_, 5-H_Ph_), 7.45 (d, 2H, *J*_2,3_ = *J*_6,5_ = 8.6 Hz, 2-H_Ph_, 6-H_Ph_) ppm. ^13^C NMR (100 MHz, CDCl_3_): *δ* = -5.2 (CH_3_Si), -5.1 (CH_3_Si), -4.7 (CH_3_Si), -4.6 (CH_3_Si), 18.2 (C_q_), 18.6 (C_q_), 25.9 (3 × CH_3_), 26.2 (3 × CH_3_), 55.6 (OCH_3_), 62.3 (C-5′), 77.1 (C-3′), 86.1 (C-4′), 91.0 (C-2′), 101.0 (C-1′), 115.2 (2 × C_Ph_), 117.4 (C_q_), 136.8 (2 × C_Ph_), 161.2 (C-O), 169.9 (N=C-S) ppm. IR (NEAT): *ν* = 2953 (C-H_Al_), 1594, 1249, 1107, 1001, 837 (C-O, C-N, C=C, C=N) cm^−1^. MS (IS): *m/z* = 526.5 [M + H]^+^. HRMS (ESI): *m*/*z* [M + H]^+^ calcd. for C_25_H_44_NO_5_SSi_2_: 526.24732; found: 526.24786.

#### 3.8.3. 2-[(3-Methoxyphenyl)sulfanyl]-4,5-dihydro(3′,5′-di-O-tert-butyldimethylsilyl-1′,2′-dideoxy-β-D-arabinofuranoso)-[1,2-d]-oxazole (**13**)

Prepared from **6** (100 mg, 0.24 mmol) and 3-iodoanisole (0.043 mL, 0.36 mmol), the obtained residue was purified by flash chromatography (eluent: petroleum ether/EtOAc, 9:1, *R_f_* = 0.25) to give **13** (51 mg, 41%) as a colorless oil; ${\left[\alpha \right]}_{D}^{20}$ = −57 (*c* = 0.95, CHCl_3_). ^1^H NMR (400 MHz, CDCl_3_): *δ* = 0.06 (s, 3H, CH_3_Si), 0.07 (s, 3H, CH_3_Si), 0.07 (s, 3H, CH_3_Si), 0.08 (s, 3H, CH_3_Si), 0.86 (s, 9H, 3×CH_3_), 0.89 (s, 9H, 3×CH_3_), 3.43 (dd, 1H, *J*_5__′b,4__′_ = 8.3 Hz, *J*_5__′b,5__′a_ = 10.5 Hz, 5′b-H), 3.63 (dd, 1H, *J*_5__′a,4__′_ = 4.6 Hz, *J*_5__′a,5__′b_ = 10.6 Hz, 5′a-H), 3.78 (s, 3H, OCH_3_), 3.87–3.91 (m, 1H, 4′-H), 4.37 (br s, 1H, 3′-H), 4.72 (dd, 1H, *J*_2__′,3__′_ = 1.2 Hz, *J*_2__′,1__′_ = 6.0 Hz, 2′-H), 5.99 (d, 1H, *J*_1__′,2__′_ = 6.0 Hz, 1′-H), 6.92 (dd, 1H, *J*_4,2_ = 2.5 Hz, *J*_4,5_ = 8.3 Hz, 4-H), 7.10 (br s, 1H, 2-H), 7.14 (d, 1H, *J*_6,5_ = 7.7 Hz, 6-H), 7.25–7.28 (m, 1H, 5-H) ppm. ^13^C NMR (100 MHz, CDCl_3_): *δ* = -5.2 (CH_3_Si), -5.1 (CH_3_Si), -4.7 (CH_3_Si), -4.6 (CH_3_Si), 18.2 (C_q_), 18.6 (C_q_), 26.0 (3 × CH_3_), 26.2 (3 × CH_3_), 55.6 (OCH_3_), 62.3 (C-5′), 77.1 (C-3′), 86.1 (C-4′), 90.9 (C-2′), 101.0 (C-1′), 116.1 (C-4), 120.1 (C-2), 127.1 (C-6), 127.8 (C_q_), 130.3 (C-5), 160.0 (C-3), 168.9 (N=C-S) ppm. IR (NEAT): *ν* = 2953 (C-H_Al_), 1591, 1250, 1106, 1002, 837 (C-O, C-N, C=C, C=N) cm^−1^. MS (IS): *m/z* = 526.5 [M + H]^+^. HRMS (ESI): *m*/*z* [M + H]^+^ calcd. for C_25_H_44_NO_5_SSi_2_: 526.24732; found: 526.24783.

#### 3.8.4. 2-[(2-Methoxyphenyl)sulfanyl]-4,5-dihydro(3′,5′-di-O-tert-butyldimethylsilyl-1′,2′-dideoxy-β-D-arabinofuranoso)-[1,2-d]-oxazole (**14**)

Prepared from **6** (90 mg, 0.21 mmol) and 2-iodoanisole (0.04 mL, 0.321 mmol), the obtained residue was purified by flash chromatography (eluent: petroleum ether/EtOAc, 85/:15, *R_f_* = 0.25) to give **14** (61 mg, 54%) as a white solid; mp 82−84 °C; ${\left[\alpha \right]}_{D}^{20}$ = −39 (*c* = 1.0, CHCl_3_). ^1^H NMR (400 MHz, CDCl_3_): *δ* = 0.06 (s, 9H, 3 × CH_3_Si), 0.07 (s, 3H, CH_3_Si), 0.86 (s, 9H, 3 × CH_3_), 0.89 (s, 9H, 3 × CH_3_), 3.45 (dd, 1H, *J*_5__′b,4__′_ = 9.1 Hz, *J*_5__′b,5__′a_ = 10.3 Hz, 5′b-H), 3.63 (dd, 1H, *J*_5__′a,4__′_ = 4.8 Hz, *J*_5__′a,5__′b_ = 10.6 Hz, 5′a-H), 3.79 (s, 3H, OCH_3_), 3.88−3.91 (m, 1H, 4′-H), 4.35 (br s, 1H, 3′-H), 4.67 (d, 1H, *J*_2__′,1__′_ = 5.9 Hz, 2′-H), 5.97 (d, 1H, *J*_1__′,2__′_ = 5.9 Hz, 1′-H), 6.92–6.96 (m, 2H, 3-H, 5-H), 7.39 (td, 1H, *J*_4,6_ = 1.2 Hz, *J*_4,3_ = *J*_4,5_ = 7.8 Hz, 4-H), 7.54 (dd, 1H, *J*_6,4_ = 1.4 Hz, *J*_6,5_ = 7.5 Hz, 6-H) ppm. ^13^C NMR (100 MHz, CDCl_3_): *δ* = -5.1 (CH_3_Si), -5.1 (CH_3_Si), -4.7 (CH_3_Si), -4.6 (CH_3_Si), 18.2 (C_q_), 18.6 (C_q_), 25.9 (3 × CH_3_), 26.2 (3 × CH_3_), 56.2 (OCH_3_), 62.5 (C-5′), 77.2 (C-3′), 86.4 (C-4′), 90.8 (C-2′), 101.2 (C-1′), 111.9 (CH_Ph_), 115.1 (C_q_), 121.3 (CH_Ph_), 132.2 (CH_Ph_), 137.0 (CH_Ph_), 159.7 (C-O), 168.7 (N=C-S) ppm. IR (NEAT): *ν* = 2930 (C-H_Al_), 1604, 1462, 1249, 1105, 1060, 1004, 814 (C-O, C-N, C=C, C=N) cm^−1^. MS (IS): *m/z* = 526.5 [M + H]^+^. HRMS (ESI): *m*/*z* [M + H]^+^ calcd. for C_25_H_44_NO_5_SSi_2_: 526.24732; found: 526.24789.

#### 3.8.5. 2-[(4-Nitrophenyl)sulfanyl]-4,5-dihydro(3′,5′-di-O-tert-butyldimethylsilyl-1′,2′-dideoxy-β-D-arabinofuranoso)-[1,2-d]-oxazole (**15**)

Prepared from **6** (100 mg, 0.24 mmol) and 1-iodo-4-nitrobenzene (89 mg, 0.36 mmol), the obtained residue was purified by flash chromatography (eluent: petroleum ether/EtOAc, 95:5, *R_f_* = 0.15) to give **15** (57 mg, 44%) as a yellow oil; ${\left[\alpha \right]}_{D}^{20}$ = −88 (*c* = 0.5, CHCl_3_). ^1^H NMR (400 MHz, CDCl_3_): *δ* = 0.04 (s, 6H, 2×CH_3_Si), 0.08 (s, 3H, CH_3_Si), 0.09 (s, 3H, CH_3_Si), 0.87 (s, 9H, 3×CH_3_), 0.88 (s, 9H, 3×CH_3_), 3.44 (dd, 1H, *J*_5__′b,4__′_ = 7.6 Hz, *J*_5__′b,5__′a_ = 10.7 Hz, 5′b-H), 3.63 (dd, 1H, *J*_5__′a,4__′_ = 4.4 Hz, *J*_5__′a,5__′b_ = 10.7 Hz, 5′a-H), 3.87-3.90 (m, 1H, 4′-H), 4.39 (d, 1H, *J*_3__′,2__′_ = 1.4 Hz, 3′-H), 4.78 (dd, 1H, *J*_2__′,3__′_ = 1.4 Hz, *J*_2__′,1__′_ = 6.0 Hz, 2′-H), 5.99 (d, 1H, *J*_1__′,2__′_ = 6.0 Hz, 1′-H), 7.79 (d, 2H, *J*_2,3_ = *J*_6,5_ = 8.6 Hz, 2-H, 6-H), 8.20 (d, 2H, *J*_3,2_ = *J*_5,6_ = 8.6 Hz, 3-H, 5-H) ppm. ^13^C NMR (100 MHz, CDCl_3_): *δ* = -5.2 (CH_3_Si), -5.1 (CH_3_Si), -4.7 (2 × CH_3_Si), 18.2 (C_q_), 18.6 (C_q_), 25.9 (3 × CH_3_), 26.1 (3 × CH_3_), 62.0 (C-5′), 76.8 (C-3′), 85.9 (C-4′), 91.3 (C-2′), 100.6 (C-1′), 124.3 (C-3, C-5), 134.3 (C-2, C-6), 135.9 (C_q_), 148.3 (C_q_), 166.7 (N=C-S) ppm. IR (NEAT): *ν* = 2953 (C-H_Al_), 1522 (N-O), 1343 (N-O), 1253, 1108, 1060, 1004, 816 (C-O, C-N, C=C, C=N) cm^−1^. MS (IS): *m/z* = 541.5 [M+H]^+^. HRMS (ESI): *m*/*z* [M + H]^+^ calcd. for C_24_H_41_N_2_O_6_SSi_2_: 541.22184; found: 541.22229. HRMS (ESI): *m*/*z* [M + Na]^+^ calcd. for C_24_H_40_N_2_NaO_6_SSi_2_: 563.20378; found: 563.20394.

#### 3.8.6. 2-[(3-Nitrophenyl)sulfanyl]-4,5-dihydro(3′,5′-di-O-tert-butyldimethylsilyl-1′,2′-dideoxy-β-D-arabinofuranoso)-[1,2-d]-oxazole (**16**)

Prepared from **6** (90 mg, 0.21 mmol) and 1-iodo-3-nitrobenzene (80 mg, 0.32 mmol), the obtained residue was purified by flash chromatography (eluent: petroleum ether/EtOAc, 9:1, *R_f_* = 0.3) to give **16** (58 mg, 50%) as a white solid; mp 85−86 °C; ${\left[\alpha \right]}_{D}^{20}$ = −74 (*c* = 1.0, CHCl_3_). ^1^H NMR (400 MHz, CDCl_3_): *δ* = 0.05 (s, 3H, CH_3_Si), 0.06 (s, 3H, CH_3_Si), 0.08 (s, 3H, CH_3_Si), 0.08 (s, 3H, CH_3_Si), 0.87 (s, 9H, 3 × CH_3_), 0.89 (s, 9H, 3 × CH_3_), 3.45 (dd, 1H, *J*_5__′b,4__′_ = 7.6 Hz, *J*_5__′b,5__′a_ = 10.7 Hz, 5′b-H), 3.64 (dd, 1H, *J*_5__′a,4__′_ = 4.4 Hz, *J*_5__′a,5__′b_ = 10.7 Hz, 5′a-H), 3.85-3.89 (m, 1H, 4′-H), 4.38 (d, 1H, *J*_3__′,2__′_ = 1.7 Hz, 3′-H), 4.78 (d, 1H, *J*_2__′,1__′_ = 6.0 Hz, 2′-H), 5.97 (d, 1H, *J*_1__′,2__′_ = 6.0 Hz, 1′-H), 7.57 (t, 1H, *J* = 8.0 Hz, 5-H), 7.91 (d, 1H, *J*_6,5_ = 7.8 Hz, 6-H), 8.24-8.26 (m, 1H, 4-H), 8.44 (s, 1H, 2-H) ppm. ^13^C NMR (100 MHz, CDCl_3_): *δ* = -5.2 (CH_3_Si), -5.1 (CH_3_Si), -4.7 (2 × CH_3_Si), 18.2 (C_q_), 18.6 (C_q_), 25.9 (3 × CH_3_), 26.1 (3 × CH_3_), 62.1 (C-5′), 76.9 (C-3′), 85.9 (C-4′), 91.5 (C-2′), 100.6 (C-1′), 124.8 (C-4), 129.5 (C-2), 129.7 (C_q_), 130.3 (C-5), 140.5 (C-6), 148.6 (C_q_), 167.4 (N=C-S) ppm. IR (NEAT): *ν* = 2930 (C-H_Al_), 1533 (N-O), 1347 (N-O), 1255, 1114, 1061, 1000, 814 (C-O, C-N, C=C, C=N) cm^−1^. MS (IS): *m/z* = 541.5 [M + H]^+^. HRMS (ESI): *m*/*z* [M + H]^+^ calcd. for C_24_H_41_N_2_O_6_SSi_2_: 541.22184; found: 541.22241. HRMS (ESI): *m*/*z* [M + Na]^+^ calcd. for C_24_H_40_N_2_NaO_6_SSi_2_: 563.20378; found: 563.20412.

#### 3.8.7. 2-[(2-Nitrophenyl)sulfanyl]-4,5-dihydro(3′,5′-di-O-tert-butyldimethylsilyl-1′,2′-dideoxy-β-D-arabinofuranoso)-[1,2-d]-oxazole (**17**)

Prepared from **6** (90 mg, 0.21 mmol) and 1-iodo-2-nitrobenzene (80 mg, 0.32 mmol), the obtained residue was purified by flash chromatography (eluent: petroleum ether/EtOAc, 9:1, *R_f_* = 0.18) to give **17** (83 mg, 72%) as a yellow solid; mp 111–113 °C; ${\left[\alpha \right]}_{D}^{20}$ = −74 (*c* = 0.5, CHCl_3_). ^1^H NMR (400 MHz, CDCl_3_): *δ* = 0.03 (s, 6H, 2 × CH_3_Si), 0.08 (s, 3H, CH_3_Si), 0.09 (s, 3H, CH_3_Si), 0.86 (s, 9H, 3×CH_3_), 0.87 (s, 9H, 3×CH_3_), 3.42 (dd, 1H, *J*_5__′b,4__′_ = 7.9 Hz, *J*_5__′b,5__′a_ = 10.7 Hz, 5′b-H), 3.62 (dd, 1H, *J*_5__′a,4__′_ = 4.6 Hz, *J*_5__′a,5__′b_ = 10.7 Hz, 5′a-H), 3.87−3.91 (m, 1H, 4′-H), 4.37 (d, 1H, *J*_3__′,4__′_ = 1.6 Hz, 3′-H), 4.76 (dd, 1H, *J*_2__′,3__′_ = 1.2 Hz, *J*_2__′,1__′_ = 6.0 Hz, 2′-H), 6.01 (d, 1H, *J*_1__′’,2__′_ = 6.0 Hz, 1′-H), 7.48−7.52 (m, 1H, 4-H), 7.58−7.62 (m, 1H, 5-H), 8.02 (d, 2H, *J* = 8.0 Hz, H-3, 6-H) ppm. ^13^C NMR (100 MHz, CDCl_3_): *δ* = -5.2 (CH_3_Si), -5.2 (CH_3_Si), -4.7 (2 × CH_3_Si), 18.2 (C_q_), 18.5 (C_q_), 25.9 (3 × CH_3_), 26.1 (3 × CH_3_), 62.3 (C-5′), 77.0 (C-3′), 86.2 (C-4′), 91.0 (C-2′), 100.8 (C-1′), 124.6 (C_q_), 125.5 (C-3), 129.8 (C-4), 133.3 (C-5), 136.0 (C-6), 150.2 (C_q_), 166.7 (N=C-S) ppm. IR (NEAT): *ν* = 2930 (C-H_Al_), 1525 (N-O), 1343 (N-O), 1254, 1109, 1064, 815 (C-O, C-N, C=C, C=N) cm^−1^. MS (IS): *m/z* = 541.5 [M + H]^+^. HRMS (ESI): *m*/*z* [M + H]^+^ calcd. for C_24_H_41_N_2_O_6_SSi_2_: 541.22184; found: 541.22238. HRMS (ESI): *m*/*z* [M + Na]^+^ calcd. for C_24_H_40_N_2_NaO_6_SSi_2_: 563.20378; found: 563.20400.

#### 3.8.8. 2-[(2-Fluorophenyl)sulfanyl]-4,5-dihydro(3′,5′-di-O-tert-butyldimethylsilyl-1′,2′-di-deoxy-β-D-arabinofuranoso)-[1,2-d]-oxazole (**18**)

Prepared from **6** (90 mg, 0.21 mmol) and 2-fluoroiodobenzene (0.04 mL, 0.32 mmol), the obtained residue was purified by flash chromatography (eluent: petroleum ether/EtOAc, 95:5, till 9:1, *R_f_* = 0.30) to give **18** (24 mg, 22%) as a colorless oil; ${\left[\alpha \right]}_{D}^{20}$ = −46 (*c* = 0.85, CHCl_3_). ^1^H NMR (400 MHz, CDCl_3_): *δ* = 0.06 (s, 3H, CH_3_Si), 0.07 (s, 9H, 3×CH_3_Si), 0.86 (s, 9H, 3 × CH_3_), 0.90 (s, 9H, 3 × CH_3_), 3.41−3.46 (m, 1H, 5′b-H), 3.61 (dd, 1H, *J*_5__′a,4__′_ = 4.5 Hz, *J*_5__′a,5__′b_ = 10.6 Hz, 5′a-H), 3.89−3.92 (m, 1H, 4′-H), 4.38 (s, 1H, 3′-H), 4.73 (d, 1H, *J*_2__′,1__′_ = 6.0 Hz, 2′-H), 5.98 (d, 1H, *J*_1__′,2__′_ = 6.0 Hz, 1′-H), 7.13−7.17 (m, 2H, 3-H, 6-H), 7.40–7.45 (m, 1H, 4-H), 7.54−7.57 (m, 1H, 5-H) ppm. ^13^C NMR (100 MHz, CDCl_3_): *δ* = -5.2 (CH_3_Si), -5.2 (CH_3_Si), -4.7 (CH_3_Si), -4.6 (CH_3_Si), 18.2 (C_q_), 18.6 (C_q_), 25.9 (3 × CH_3_), 26.2 (3 × CH_3_), 62.3 (C-5′), 77.1 (C-3′), 86.4 (C-4′), 91.3 (C-2′), 101.1 (C-1′), 114.4 (d, ^2^*J*_C-F_ = 18.4 Hz, C-1), 116.70 (d, ^2^*J*_C-F_ = 22.5 Hz, C-3), 125.0 (d, ^3^*J*_C-F_ = 3.9 Hz, C-6), 132.9 (d, ^3^*J*_C-F_ = 8.1 Hz, C-4), 137.0 (C-5), 162.7 (d, ^1^*J*_C-F_ = 251.0 Hz, C-2), 167.5 (N=C-S) ppm. ^19^F NMR (376 MHz, CDCl_3_): *δ* = -104.91 (s, F) ppm. IR (NEAT): *ν* = 2953, 2930 (C-H_Al_), 1606, 1475, 1255, 1107 (C-F), 1062, 1001, 815 (C-O, C-N, C=C, C=N) cm^−1^. MS (IS): *m/z* = 514.5 [M + H]^+^. HRMS (ESI): *m*/*z* [M + H]^+^ calcd. for C_24_H_41_FNO_4_SSi_2_: 514.22734; found: 514.22782. HRMS (ESI): *m*/*z* [M + Na]^+^ calcd. for C_24_H_40_ FNNaO_4_SSi_2_: 536.20928; found: 536.20943.

#### 3.8.9. 2-[(2-Trifluoromethylphenyl)sulfanyl]-4,5-dihydro(3′,5′-di-O-tert-butyldimethyl-silyl-1′,2′-dideoxy-β-D-arabinofuranoso)-[1,2-d]-oxazole (**19**)

Prepared from **6** (90 mg, 0.21 mmol) and *o*-iodotrifluoromethylbenzene (0.05 mL, 0.32 mmol), the obtained residue was purified by flash chromatography (eluent: petroleum ether/EtOAc, 9:1, *R_f_* = 0.27) to give **19** (40 mg, 33%) as a white solid; mp 98–100 °C; ${\left[\alpha \right]}_{D}^{20}$ = −49 (*c* = 1.04, CHCl_3_). ^1^H NMR (400 MHz, CDCl_3_): *δ* = 0.03 (s, 3H, CH_3_Si), 0.06 (s, 6H, 2x(CH_3_Si)), 0.07 (s, 3H, CH_3_Si), 0.85 (s, 9H, 3 × CH_3_), 0.89 (s, 9H, 3 × CH_3_), 3.41 (dd, 1H, *J*_5__′b,4__′_ = 8.5 Hz, *J*_5__′b, 5__′a_ = 10.6 Hz, 5′b-H), 3.61 (dd, 1H, *J*_5__′a,4__′_ = 4.6 Hz, *J*_5__′a,5__′b_ = 10.6 Hz, 5′a-H), 3.85–3.89 (m, 1H, 4′-H), 4.35 (br s, 1H, 3′-H), 4.73 (dd, 1H, *J*_2__′,3__′_ = 1.2 Hz, *J*_2__′,1__′_ = 6.0 Hz, 2′-H), 5.96 (d, 1H, *J*_1__′,2__′_ = 6.0 Hz, 1′-H), 7.51–7.58 (m, 2H, 4-H, 5-H), 7.74−7.77 (m, 2H, 3-H, 6-H) ppm. ^13^C NMR (100 MHz, CDCl_3_): *δ* = −5.3 (CH_3_Si), −5.2 (CH_3_Si), −4.7 (CH_3_Si), -4.7 (CH_3_Si), 18.2 (C_q_), 18.6 (C_q_), 25.9 (3×CH_3_), 26.1 (3×CH_3_), 62.3 (C-5′), 77.0 (C-3′), 86.2 (C-4′), 91.3 (C-2′), 100.9 (C-1′), 123.1 (q, ^1^*J*_C-F_ = 273.9 Hz, CF_3_), 125.7 (C-1), 127.4 (q, ^3^*J*_C-F_ = 5.3 Hz, C-3), 130.5 (C-4), 132.6 (C-5), 133.5 (q, ^2^*J*_C-F_ = 30.2 Hz, C-2), 139.5 (C-6), 168.0 (N=C-S) ppm. ^19^F NMR (376 MHz, CDCl_3_): *δ* = −59.87 (s, 3F) ppm. IR (NEAT): *ν* = 2930 (C-H_Al_), 1605, 1312, 1254, 1164 (C-F), 1131, 1001, 812 (C-O, C-N, C=C, C=N) cm^−1^. MS (IS): *m/z* = 564 [M + H]^+^. HRMS (ESI): *m*/*z* [M + H]^+^ calcd. for C_25_H_41_F_3_NO_4_SSi_2_: 564.22414; found: 564.22464. HRMS (ESI): *m*/*z* [M + Na]^+^ calcd. for C_25_H_40_ F_3_NNaO_4_SSi_2_: 586.20609; found: 586.20630.

#### 3.8.10. 2-[(2-Nitrophenyl)sulfanyl]-4,5-dihydro(3′,5′-di-O-tert-butyldimethylsilyl-1′,2′-dideoxy-α-D-xylofuranoso)-[1,2-d]-oxazole (**20**)

Prepared from **7** (90 mg, 0.21 mmol) and 1-iodo-2-nitrobenzene (80 mg, 0.32 mmol), the obtained residue was purified by flash chromatography (eluent: petroleum ether/EtOAc, 9:1, *R_f_* = 0.16) to give **20** (67 mg, 58%) as a yellow oil; ${\left[\alpha \right]}_{D}^{20}$ = +56 (*c* = 0.57, CHCl_3_). ^1^H NMR (400 MHz, CDCl_3_): *δ* = 0.04 (s, 3H, CH_3_Si), 0.05 (s, 3H, CH_3_Si), 0.09 (s, 3H, CH_3_Si), 0.11 (s, 3H, CH_3_Si), 0.87 (s, 9H, 3 × CH_3_), 0.88 (s, 9H, 3 × CH_3_), 3.68–3.72 (m, 1H, 4′-H), 3.76–3.84 (m, 2H, 5′-H), 4.25 (d, 1H, *J*_3__′,4__′_ = 3.1 Hz, 3′-H), 4.69 (d, 1H, *J*_2__′,1__′_ = 5.5 Hz, 2′-H), 6.06 (d, 1H, *J*_1__′,2__′_ = 5.5 Hz, 1′-H), 7.48–7.52 (m, 1H, 4-H), 7.59−7.62 (m, 1H, 5-H), 8.04 (d, 2H, *J* = 8.1 Hz, 3-H, 6-H) ppm. ^13^C NMR (100 MHz, CDCl_3_): *δ* = -5.2 (CH_3_Si), -5.1 (CH_3_Si), -4.9 (CH_3_Si), -4.9 (CH_3_Si), 18.3 (C_q_), 18.5 (C_q_), 25.9 (3 × CH_3_), 26.2 (3 × CH_3_), 60.1 (C-5′), 75.0 (C-3′), 80.3 (C-4′), 88.9 (C-2′), 100.1 (C-1′), 124.7 (C_q_), 125.7 (C-3), 129.8 (C-4), 133.4 (C-5), 136.0 (C-6), 150.2 (C_q_), 166.7 (N=C-S) ppm. IR (NEAT): *ν* = 2930 (C-H_Al_), 1528 (N-O), 1346 (N-O), 1254, 1102, 1004, 814 (C-O, C-N, C=C, C=N) cm^−1^. MS (IS): *m/z* = 541.5 [M + H]^+^. HRMS (ESI): *m*/*z* [M + H]^+^ calcd. for C_24_H_41_N_2_O_6_SSi_2_: 541.22184; found: 541.22158. HRMS (ESI): *m*/*z* [M + Na]^+^ calcd. for C_24_H_40_N_2_NaO_6_SSi_2_: 563.20378; found: 563.20327.

#### 3.8.11. 2-[(2-Methoxyphenyl)sulfanyl]-4,5-dihydro(3′,5′-di-O-tert-butyldimethylsilyl-1′,2′-dideoxy-α-D-xylofuranoso)-[1,2-d]-oxazole (**21**)

Prepared from **7** (90 mg, 0.21 mmol) and 1-iodo-2-nitrobenzene (80 mg, 0.32 mmol), the obtained residue was purified by flash chromatography (eluent: petroleum ether/EtOAc, 9:1, *R_f_* = 0.17) to give **21** (50 mg, 45%) as a colorless oil; ${\left[\alpha \right]}_{D}^{20}$ = +18 (*c* = 0.86, CHCl_3_). ^1^H NMR (400 MHz, CDCl_3_): *δ* = 0.04 (s, 3H, CH_3_Si), 0.04 (s, 3H, CH_3_Si), 0.07 (s, 3H, CH_3_Si), 0.09 (s, 3H, CH_3_Si), 0.87 (s, 18H, 6 × CH_3_), 3.69−3.73 (m, 1H, 4′-H), 3.76−3.83 (m, 2H, 5′-H), 3.86 (s, 3H, OCH_3_), 4.21 (d, 1H, *J*_3__′,4__′_ =2.9 Hz, 3′-H), 4.63 (d, 1H, *J*_2__′,1__′_ = 5.4 Hz, 2′-H), 6.00 (d, 1H, *J*_1__′,2__′_ = 5.4 Hz, 1′-H), 6.92−6.97 (m, 2H, 3-H, 5-H), 7.39−7.41 (m, 1H, 4-H), 7.57 (dd, 1H, *J* = 1.6 Hz, *J* = 7.6 Hz, 6-H) ppm. ^13^C NMR (100 MHz, CDCl_3_): *δ* = -5.2 (CH_3_Si), -5.1 (CH_3_Si), -4.9 (CH_3_Si), -4.6 (CH_3_Si), 18.3 (C_q_), 18.5 (C_q_), 25.9 (3 × CH_3_), 26.2 (3 × CH_3_), 56.2 (OCH_3_), 60.2 (C-5′), 75.1 (C-3′), 79.9 (C-4′), 88.7 (C-2′), 100.2 (C-1′), 111.9 (C-3), 115.4 (C-4), 121.4 (C_q_), 132.1 (C-5), 136.6 (C-6), 159.6 (C_q_), 168.5 (N=C-S) ppm. IR (NEAT): *ν* = 2930 (C-H_Al_), 1603, 1474, 1252, 1105, 1065, 1007, 812 (C-O, C-N, C=C, C=N) cm^−1^. MS (IS): *m/z* = 526.5 [M + H]^+^. HRMS (ESI): *m*/*z* [M + H]^+^ calcd. for C_24_H_41_N_2_O_6_SSi_2_: 526.24732; found: 526.24718.

#### 3.8.12. 2-[(2-Nitrophenyl)sulfanyl]-4,5-dihydro(3′,5′-di-O-tert-butyldimethylsilyl-1′,2′-dideoxy-β-D-ribofuranoso)-[1,2-d]-oxazole (**22**)

Prepared from **8** (120 mg, 0.28 mmol) and 1-iodo-2-nitrobenzene (107 mg, 0.43 mmol), the obtained residue was purified by flash chromatography (eluent: petroleum ether/EtOAc, 9:1, *R_f_* = 0.16) to give **22** (53 mg, 46%) as a yellowish solid. ^1^H NMR (400 MHz, CDCl_3_): *δ* = 0.04 (s, 3H, CH_3_Si), 0.04 (s, 3H, CH_3_Si), 0.08 (s, 3H, CH_3_Si), 0.11 (s, 3H, CH_3_Si), 0.87 (s, 9H, 3 × CH_3_), 0.88 (s, 9H, 3 × CH_3_), 3.67-3.70 (m, 1H, 4′-H), 3.75-3.83 (m, 2H, 5′-H), 4.26 (d, 1H, *J* = 3.2 Hz, 3′-H), 4.73 (d, 1H, *J* = 5.5 Hz, 2′-H), 6.03 (d, 1H, *J* = 5.5 Hz, 1′-H), 7.58 (t, 1H, *J* = 8.0 Hz, 5-H), 7.96 (d, 1H, *J* = 7.8 Hz 6-H), 8.24 (d, 1H, *J* = 7.7 Hz, 4-H), 8.46 (br s, 1H, 2-H) ppm. ^13^C NMR (100 MHz, CDCl_3_): *δ* = -5.2 (CH_3_Si), -5.1 (CH_3_Si), -4.9 (CH_3_Si), -4.6 (CH_3_Si), 18.3 (C_q_), 18.5 (C_q_), 25.9 (3 × CH_3_), 26.2 (3 × CH_3_), 60.2 (C-5′), 75.2 (C-3′), 80.3 (C-4′), 89.4 (C-2′), 100.0 (C-1′), 124.7 (CH), 129.4 (CH), 129.8 (C_q_), 130.3 (CH), 140.5 (CH), 148.6 (C_q_), 167.4 (N=C-S) ppm. IR (NEAT): *ν* = 2929 (C-H_Al_), 1532, 1341 (C-NO_2_), 1251, 1094, 1059, 810 (C-O, C-N, C=C, C=N) cm^−1^. MS (IS): *m/z* = 541.5 [M + H]^+^. HRMS (ESI): *m*/*z* [M + H]^+^ calcd. for C_24_H_41_N_2_O_6_SSi_2_: 541.22184; found: 541.22238.

## 4. Conclusions

In summary, our continuous efforts to study chiral 1,3-oxazolidine-2-thione anchored onto carbohydrate templates allowed us to describe the application of a copper-catalyzed carbon–sulfur bond formation. Despite a few yield limitations, this methodology provided an alternative chemical tool for 1,3-oxazolidine-2-thione functionalization, and to the best of our knowledge, a unique method to link aromatic rings onto chiral thioamide-derived heterocycles. The structures of all of the synthesized compounds were confirmed by detailed NMR spectroscopy and HRMS investigations. A further investigation to broaden the scope of the reactions and their selectivity to other templates as well as coupling reactions is currently being undertaken.

## Data Availability

The data presented in this study are available on request from the corresponding authors.

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
