# Peer review of "Copper-catalyzed S-arylation of Furanose-Fused Oxazolidine-2-thiones"

_molecules, 2022, doi:10.3390/molecules27175597_

Round 1
Reviewer 1 Report
Authors synthesized carbohydrate-fused oxazolidine-2-thiones by using copper catalyzed S-arylation method. The current study is interesting; however, the authors should address the following comments to improve the quality of the manuscript:
1. Title needs to be changed from ‘‘carbohydrate-fused’’ to furanosides specifically.
2. Abstract is short it should be elaborated to understand the theme of manuscript.
3. Introduction line 15 “many various” either many or various.
4. Introduction is very short; it should be elaborated by mentioning about importance of 1,2-annulated sugars.
5. Authors may comment on scope of present methodology for pyranosides.
6. Authors need to mention the temperature for optical rotation values.
7. What are the other products formed in low yield examples, 18 and 19 in particular.
8. The authors are recommended to cite following relevant articles related to importance of 1,2-annulated sugars in the introduction section;
i) Awan, S. I.; Werz, D. B. Syntheses of 1,2-annulated and 1-spiroannulated carbohydrate derivatives: Recent developments. Bioorg. Med. Chem. 2012, 20, 1846–1856. https://doi.org/10.1016/j.bmc.2011.10.089
ii) Leibeling, M.; Milde, B.; Kratzert, D.; Stalke, D.; Werz, D. B. Intermolecular Twofold Carbopalladation/Cyclization Sequence to Access Chromans and Isochromans from Carbohydrates. Chem. Eur. J. 2011, 17, 9888–9892. https://doi.org/10.1002/chem.201101917
iii) Chennaiah, A.; Dubbu, S.; Parasuraman, K.; Vankar, Y. D. Stereoselective Synthesis of 1,2-Annulated Sugars Having Substituted Tetrahydropyran/(-furan) Scaffolds Using the PrinsReaction. Eur. J. Org. Chem. 2018, 2018, 6706−6713. https://doi.org/10.1002/ejoc.201801273
iv) Verma, A. K.; Chennaiah, A.; Dubbu, S.; Vankar, Y. D. Palladium catalyzed synthesis of sugar-fused indolines via C(sp2)-H/N-H activation. Carbohydr. Res. 2019, 473, 57–65. https://doi.org/10.1016/j.carres.2018.12.015
Overall, after addressing the points mentioned above, I recommend this article to publish in molecules.
Author Response
We would like to thank the reviewer for the comments, which allowed us to improve the quality of the manuscript and hopefully made it acceptable for publication in Molecules. Our point-to-point responses can be found in the attached file.

Reviewer 2 Report
This paper discusses the copper-catalyzed S-arylation of carbohydrate anchored 1,3-oxazolidine-2-thiones. The reaction shows sufficient chemoselectivity in the S-arylation reaction, although the yields are moderate or good, in some cases are poor. The chemoselective S-arylation reaction catalyzed by Cu(I) salt should be worth publishing in this Journal. However, before the publication the manuscript should be revised as described below.
Page 2; The preparation of 2 and 3 should be discussed in the text, although the synthesis of 1 is described.
Table 1 and Table 2; Some of the reactions show relatively low yield. The authors must specify remaining substances, although they only specify the recovery of the molecules 9 and 10 in this reaction.
Page 6; The most efficient iodo benzene derivative with the OMe substituent should be para-derivative. This is at odds with the discussion.
Furthermore, the authors need to propose a reaction mechanism for this reaction. Why does S-arylation selectivity occur, and what are the roles of copper and ligands?
Author Response

(The authors gave the same response as above.)

Reviewer 3 Report
The authors report cross-coupling of carbohydrate-fused oxazolidine-2-thiones with aryl iodides. The formation of chemical bonds through the cross-coupling is very attractive and advantageous with respect to other methods. In the present study, the sulfur atom was selectively arylated in the presence of a nitrogen nucleophilic center. A sufficient range of products has been obtained in modest-to-good yields (22-72%). This approach is consistent, and cheap, as demonstrated with the number of products prepared, using one of the cheapest transition metals. There is a good experimental section, including the full characterization of the products, supported NMR spectra.
Therefore, I recommend publication of the manuscript in Molecules after minor revision as follows:
The authors did not propose the reaction mechanism. Particularly, in the previous study [Tetrahedron Letters 49 (2008) 5583–558] the arylation of the carbohydrate-based 1,3-oxazolidine-2-thione with aryl boronic acid led to the sulphur elimination from the product. I guess this issue would be clear to readers to follow the reaction mechanism on the scheme.
Author Response

(The authors gave the same response as above.)

Round 2
Reviewer 2 Report
This paper discusses the copper-catalyzed S-arylation of carbohydrate anchored 1,3-oxazolidine-2-thiones. The chemoselective S-arylation reaction catalyzed by Cu(I) salt should be worth publishing in this Journal. The reviewer feels the revision was properly established by reflecting the reviewer's opinion. The reviewer considers the revised manuscript to become suitable for publication. Thus, the reviewer recommends publication of this paper as this version.